# The Formation of Mn-Ce Oxide Catalysts for CO Oxidation by Oxalate Route: The Role of Manganese Content

**DOI:** 10.3390/nano11040988

**Published:** 2021-04-12

**Authors:** Olga A. Bulavchenko, Tatyana N. Afonasenko, Alexey R. Osipov, Alena A. Pochtar’, Andrey A. Saraev, Zahar S. Vinokurov, Evgeny Yu. Gerasimov, Sergey V. Tsybulya

**Affiliations:** 1Boreskov Institute of Catalysis SB RAS, Lavrentiev Ave. 5, 630090 Novosibirsk, Russia; po4tar@catalysis.ru (A.A.P.); asaraev@catalysis.ru (A.A.S.); vinzux@mail.ru (Z.S.V.); gerasimov@catalysis.ru (E.Y.G.); tsybulya@catalysis.ru (S.V.T.); 2Center of New Chemical Technologies BIC, Boreskov Institute of Catalysis, Neftezavodskaya 54, 644040 Omsk, Russia; atnik@ihcp.ru (T.N.A.); dysprozii666@gmail.com (A.R.O.)

**Keywords:** manganese oxide, ceria, solid solution, nanostructure, CO oxidation

## Abstract

The Mn-Ce oxide catalysts active in the oxidation of CO were studied by X-ray diffraction (XRD), X-ray photoelectron spectroscopy (XPS), temperature-programmed reduction (TPR), transition electron microscopy (TEM), energy dispersive X-Ray (EDX), and a differential dissolution technique. The Mn-Ce catalysts were prepared by thermal decomposition of oxalates by varying the Mn:Ce ratio. The nanocrystalline oxides with a fluorite structure and particle sizes of 4–6 nm were formed. The introduction of manganese led to a reduction of the oxide particle size, a decrease in the surface area, and the formation of a Mn_y_Ce_1−y_O_2−δ_ solid solution. An increase in the manganese content resulted in the formation of manganese oxides such as Mn_2_O_3_, Mn_3_O_4_, and Mn_5_O_8_. The catalytic activity as a function of the manganese content had a volcano-like shape. The best catalytic performance was exhibited by the catalyst containing ca. 50 at.% Mn due to the high specific surface area, the formation of the solid solution, and the maximum content of the solid solution.

## 1. Introduction

The combustion of fossil fuels in power plants, motor vehicles, and other industries emits different toxic pollutants such as CO, NO_x_, and volatile organic compounds (VOCs), which are very dangerous for the environment and human health. In particular, emissions of carbon monoxide pose a serious threat to human health and ecology. The elimination of CO via catalytic oxidation is one of the most effective methods for the purification of exhaust gases from automobiles and industry [1,2]. For these purposes, manganese-containing catalysts have received considerable attention [3,4]. They are an alternative to noble metal catalysts because, along with their low cost, they have thermal stability and resistance to sulfur poisoning [5]. The catalytic activity in oxidation reactions on manganese oxides is associated with their structural flexibility and variability of the oxidation state of manganese [3]. 

Ceria (CeO_2_) has inferior activity in the oxidation reactions of CO, but it has been widely applied as a catalyst support or a surface modifier [6]. For example, CeO_2_ is well known as an irreplaceable component in three-way catalysts [7,8] and low-temperature selective catalytic reduction of NO_x_ [9]. The incredible performance of ceria has been traditionally associated with the excellent ability of these oxides to shuttle between Ce(III) and Ce(IV) states, their oxygen storage capacity, and abundant oxygen vacancies [6]. 

The introduction of a redox-active cation into ceria to form a solid solution or a strong metal−support interaction changes the properties of materials. The cooperation of manganese and cerium oxides enhances oxygen mobility and redox ability [10,11,12]. The Mn-Ce oxides show excellent catalytic performance in the oxidation of hydrocarbons, volatile organic compounds, CO [10,11,12,13,14,15,16,17,18,19,20] and selective catalytic oxidation of NO_x_ with NH_3_ [5,21]. Recently, the number of works devoted to Mn-Ce oxide catalysts has sharply increased [12,15,16,17,22,23,24,25,26,27]; special attention is paid to improving the synthesis methods leading to the formation of active, highly dispersed and nanostructured compounds. For the preparation of MnO_x_-CeO_2_ catalysts, methods such as combustion [11], coprecipitation [19,28,29], sol–gel [19,25], and template synthesis [14] are used. Saker et al. [30] employed the impregnation of a MnO_x_-CeO_2_ catalyst on Al_2_O_3_. Delimas et al. [11] used urea combustion to prepare a Mn_0.5_Ce_0.5_O_2_ catalyst. Du et al. [12] applied hydrothermal synthesis combined with redox deposition.

Three kinds of active states existed in the MnO_x_-CeO_2_ catalysts: (i) Mn cations in the fluorite structure, (ii) active dispersed MnO_x_ surface particles strongly interacting with CeO_2_ oxide, and (iii) relatively large MnO_x_ aggregates on the surface of the support [31]. Depending on the preparation conditions, one of these states prevails.

The controlled thermal decomposition of manganese or cerium oxalates separately has been extensively studied, and the formation of highly dispersed oxides with a high surface area has been shown [32,33,34,35,36,37]. Much less attention is paid to the formation of double Mn-Ce oxides. Li et al. [38] studied the effect of the calcination temperature of oxalates on the properties of Mn-Ce oxide catalysts supported on SiO_2_. For such systems, the low synthesis temperature leads to low crystallinity. The oxalate route is commonly used to prepare oxide catalysts, since this method is simple, provides the ability to control the properties of the product by varying the decomposition conditions, has a relatively low decomposition temperature and the ability to obtain a product with a high specific surface area. 

Previously, we studied the influence of the annealing conditions on the catalytic, structural, microstructural and redox properties of the Mn-Ce oxide catalysts obtained by thermal decomposition of oxalate precursors [39]. The goal of the current study is to determine the effect of Mn content in Mn-Ce oxide catalysts on the catalyst properties. For this, a series of *MnxCe1−x* oxides were synthesized by varying the x content from 0 to 1 and studied by a complex of physicochemical methods, including ex situ and in situ X-ray diffraction (XRD), high-resolution transmission electron microscopy (TEM) with energy dispersive X-ray analysis (EDX), temperature-programmed reduction (TPR-H_2_), X-ray photoelectron spectroscopy (XPS), and a differential dissolution (DD) technique; the catalytic activity was tested in the reaction of CO oxidation.

## 2. Materials and Methods

### 2.1. Catalysts Preparation

Precipitation was carried out with vigorous stirring of a mixed Mn(NO_3_)_2_ and Ce(NO_3_)_3_ solution by the dropwise addition of a (NH_4_)_2_C_2_O_4_ solution. The deposition time was 1 h. After adding the entire volume of precipitant, stirring of the solution with the precipitate formed continued for 1 h. The resulting precipitate was filtered off and washed with water (~2 L) on a filter. Then, it was dried at 60 °C for 20 h and calcined at 400 °C for 4 h. The resulting catalysts were designated as *MnxCe1−x*, where x is the molar fraction of manganese.

### 2.2. Catalyst Characterization

The elemental composition of samples was determined using an X-ray fluorescence spectrometer ARL Advant’X 2247 (Thermo Fisher Scientific, Waltham, MA, USA) equipped with Rh anode as an X-ray source. The mass percentage of elements was estimated using QuantAS software.

The phase composition was determined by XRD using a Bruker D8 Advance (Germany) diffractometer equipped with a Lynxeye linear detector. The diffraction patterns were obtained in the 2θ range from 15 to 85° with a step of 0.05° using monochromatic Cu K_α_ radiation (λ = 1.5418 Å). Rietveld refinement for quantitative analysis was carried out using the software package Topas V.4.2. The instrumental broadening was described with metallic silicon as reference material. The size of the coherent scattering domain was calculated using LVol-IB values (LVol-IB—the volume-weighted mean column height based on integral breadth). In situ XRD experiments were performed at the Precision Diffractometry-2 station at the Siberian Synchrotron and Terahertz Radiation Centre (Institute of Nuclear Physics SB RAS, Novosibirsk). The station was equipped with a single-coordinate position-sensitive detector OD-3M (Institute of Nuclear Physics SB RAS); a high-temperature X-ray chamber-reactor XRK 900 (Anton Paar GmbH, Graz, Austria), and a gas mixture feed system equipped with a digital mass flow controllers SEC-Z500 (Horiba Ltd., Kyoto, Japan). The radiation wavelength of 0.17238 nm was set by a single reflection of the incoming white synchrotron radiation beam from a Ge (111) crystal. High-temperature in situ XRD measurements for the *Mn0.45Ce0.55* sample were carried out in a flow mode with a gas mixture of 10% H_2_ + He and a gas flow rate of 100 sccm. The sample treatment included subsequent heating from 30 to 600 °C with a rate of 5 °C/min and cooling down to 30 °C with a rate of 30 °C/min.

The specific surface area was calculated with the Brunauer–Emmett–Teller (BET) method using nitrogen adsorption isotherms measured at liquid nitrogen temperatures on an automatic Micromeritics ASAP 2400 sorptometer (Micromeritics Instrument. Corp., Norcross, GA, USA).

The differential dissolution method was used to determine the cationic composition of the various phases of the oxides. Differential dissolution was carried out in a dynamic mode using a stoichiograph equipped with an inductively coupled plasma atomic emission spectrometer (BAIRD). The fraction with the average particle size ≤ 40 μm was analyzed. A ca. 3 mg weighed portion of the sample was supported homogeneously onto an expendable tape made of a polymer film with an adhesive surface. The tape was then placed in the flow microreactor of the stoichiograph. Using the stoichiographic titration mode [40], dissolution was started with an aqueous solution of HCl (1.2 M) and then successively proceeded to HCl (3 M) and HF (3.8 M) solutions. The complete dissolution of the sample occurred in the HF (3.8 M) solution. The composition of the samples was determined using spectral lines of the elements: 413.6 nm for Ce and 293.3 nm for Mn with a 5% accuracy of measurements at a sensitivity of 10^−3^ μg mL^−1^. Since oxygen cannot be determined by this method, stoichiometric formulas of the corresponding phases are presented in their fragmentary form without oxygen. The calculation method of DD data is given in Supporting Information.

The morphology and microstructure of the catalysts were studied by transmission electron microscopy. TEM images were obtained using a JEM-2010 (JEOL, Tokyo, Japan) and ThemisZ microscope (Thermo Fisher Scientific, Eindhoven, Netherlands) with a resolution of 1.4 Å and 0.7 Å, respectively. Point EDX analysis was carried out using energy dispersive spectrometer XFlash Bruker with an energy resolution of 128 eV. Elemental maps were obtained using energy dispersive spectrometer SuperX Thermo Fisher Scientific. Samples for research were fixed on standard copper grids using ultrasonic dispersion of the catalysts in ethanol.

The temperature-programmed reduction in hydrogen (TPR-H_2_) was performed with 40–60 mg of sample in a quartz reactor using a flow setup with a thermal conductivity detector. The reducing mixture (10 vol. % of H_2_ in Ar) was fed at 40 mL/min. The rate of heating from room temperature to 700 °C was 10 °C/min. The TPR curves were normalized per catalyst mass. 

The XPS measurements were performed on a photoelectron spectrometer (SPECS Surface Nano Analysis GmbH, Berlin, Germany) equipped with a PHOIBOS-150 hemispherical electron energy analyzer and an XR-50 X-ray source with a double Al/Mg anode. The core-level spectra were obtained using AlKα radiation (hν = 1486.6 eV) under ultrahigh vacuum conditions. The binding energy (E_b_) of photoemission peaks was corrected for the Ce3d_3/2_-U’’’ peak (E_b_ = 916.7 eV) of cerium oxide. The curve fitting was done by the CasaXPS software [41]. The line shape used in the fit was the sum of Lorentzian and Gaussian functions. A Shirley-type background was subtracted from each spectrum [42]. Relative element concentrations were determined from the integral intensities of the core-level spectra using the theoretical photoionization cross-sections according to Scofield [43].

### 2.3. Catalytic Tests

Catalytic tests in the reaction of CO oxidation were performed in a flow regime in a glass reactor having 170 mm in length and 10 mm in the inner diameter. The initial gas mixture composition was 1 vol.% CO in air. All the samples were investigated in the temperature range of 50–400 °C. The contact time was 0.06 s. A catalyst fraction of 0.4–1.0 mm was used, a catalyst mass was 0.5 g. To avoid overheating during the exothermic reaction, the catalyst was mixed with a quartz powder of the same particle size. The reactant mixture before and after the reactor was analyzed by a gas chromatograph equipped with a zeolite CaA column and a thermal conductivity detector. 

The catalytic activity was calculated from the formula: R(CO) = C_0_ X·V/m_cat_, [cm^3^(CO)/g∙s]
X = (P_0_ − P_cur_)/P_0_
where P_0_ is the peak area corresponding to the initial concentration of CO in the reactant mixture; P_cur_ is the peak area corresponding to the current concentration of CO at the reactor outlet; X is the degree of CO conversion; C_0_ is the initial concentration of CO in the mixture (C_0_ = 1 vol.%); V is the feed rate of the reactant mixture, mL/min; and m_cat_ is the mass of the catalyst, g. 

## 3. Results

### 3.1. Structural Properties 

Figure 1 shows the diffraction patterns of the *MnxCe1−x* catalysts with x varying from 0 to 1. 

The XRD patterns contain wide peaks located at 2θ = 32.8, 47.4, 56.4, 59.6, 69.5, 76.9, 79.1° and correspond to reflections of CeO_2_ with the fluorite-type structure, space group *Fm3m* [JCPDS No. 431002]. An increase in the Mn content to x = 0.15–0.45 leads to the appearance of additional reflections located at 2θ = 18.2, 21.6, 36.2, 38.4, 64.8, 66.0º. These peaks belong to manganese oxides Mn_3_O_4_ [JCPDS No. 240734], Mn_2_O_3_ [JCPDS No. 411442], and Mn_5_O_8_ [JCPDS No. 200718]. The intensity of reflections of manganese oxides grows with increasing Mn content. The phase content was estimated by the Rietveld refinement and summarized in Table 1. From x = 0.25 to 0.9, the fraction of simple Mn oxides rises from 4 to 80 wt.%. The *Mn1Ce0* catalyst contains only Mn oxides in different modifications: 28 wt.% Mn_2_O_3_, 34 wt.% Mn_3_O_4_, and 38 wt.% Mn_5_O_8_. Such a variety of valence states of manganese after calcination at 400 °C is probably due to the exothermic effect during the calcination of manganese oxalate in the air.

The evolution of the lattice parameter and crystallite size of ceria is presented in Figure 2. In the case of pure ceria for *Mn0Ce1*, the lattice parameter is 5.412 Å, whereas for *MnxCe1−x*, the value varies in the range of 5.412–5.402 Å (Figure 2, Table 1). For x = 0–0.25, the crystallite size and lattice parameter hardly change. A further increase in the Mn content to x = 0.45–0.7 results in a decrease in the crystallite size and lattice parameter. For x = 0.8–0.9, it is difficult to correctly estimate the structural characteristics of ceria due to low intensities of their diffraction lines and overlap with reflections of manganese oxide. The surface area of the oxides does not change significantly with varying the chemical composition from x = 0 to 0.45 and is about 99–110 m^2^/g; then the surface area gradually decreases to 32 m^2^/g for x = 1 (Table 1).

It is noteworthy that structural features of ceria in the *Mn0.45Ce0.55-Mn0.7Ce0.3* catalysts are different from other samples in the series (Figure 2). Additionally, a slight shift of the lattice parameters of ceria gives us a hint that the formation of a Mn_y_Ce_1−y_O_2−δ_ solid solution occurs. The incorporation of a smaller size Mn ion structure (Mn^2+^—0.097 nm, Mn^3+^—0.065 nm, Mn^4+^—0.053 nm, Ce^4+^—0.097 nm, Ce^3+^—0.114 nm) into the oxide results in a decrease in the lattice parameter. To confirm the distribution of cations over the oxide, we analyzed the structural and microstructural characteristics of *Mn0.45Ce0.55* in comparison with *Mn0.15Ce0.85* (*Mn0.1Ce0.9*) using such methods as DD and TEM with EDX microanalysis. DD gives information about the joint or separate dissolution of Mn and Ce cations by changing the dissolution conditions. TEM shows the microstructural characteristics of oxides and cation distribution. 

The catalysts were studied by high-resolution transmission electron microscopy coupled with energy dispersive X-ray analysis. The results for the *Mn0.15Ce0.85* catalyst are presented in Figure 3. The large plate aggregates with sizes of ca. 1000 nm are observed (Figure 3a). They consist of primary disordered particles with sizes of 5–8 nm (Figure 3b,c). The interplanar distance is 3.18Å (Figure 3c), which corresponds to the (111) plane of fluorite. EDX analysis in this area shows only Ce (Figure 3d). In addition, particles differing in morphology (separately located) are observed, for which the distance is 4.88 Å, which corresponds to the (101) plane of Mn_3_O_4_ or (200) plane of Mn_5_O_8_ (Figure 3e). EDX analysis showed that in these areas, along with manganese atoms, 3–5 at.% Ce is present (Figure 3f); however, this signal can be observed from single CeO_2_ crystallites located on the surface of massive manganese oxide particles.

In the case of catalysts with higher manganese content, *Mn0.45Ce0.55*, there are two types of particles (Figure 4a,b). Particles of the first type have rounded shapes and sizes of 10–50 nm (Figure 4c). According to EDX analysis, there is the Mn-rich area; the Mn:Ce ratio is about 98:2 (Figure 4d). The interplanar distances of 4.85 Å are observed, which correspond to the (101) plane of Mn_3_O_4_ or (200) plane of Mn_5_O_8_. The second type of particles are elongated, have a uniform shape (Figure 4a), and consist of dispersed crystallites 2–4 nm in size (Figure 4e). The interplanar distance of 3.12 Å corresponds to the (111) plane of fluorite. According to the EDX analysis, these areas contain about 17 and 83 at. % of Mn and Ce cations, respectively (Figure 4f). To confirm the distribution of elements, we applied EDS mapping (Figure 4g,h). EDS mapping from this area shows a uniform distribution of manganese and cerium cations over the particle with a cation ratio of Mn:Ce is about 20:80.

In contrast to XRD and TEM methods, which reveal only structural and microstructural changes, the differential dissolution method is able to determine the cationic composition of phases. This method determines the stoichiometry of the elemental composition of successively dissolving phases. The propagation of the dissolution reaction front from the surface to the center of the particles is accompanied by continuous recording of the ratio between all elements. 

For the *Mn0.45Ce0.55* catalyst, Figure 5 shows the dissolution curves for manganese and cerium depending on time, as well as the Mn:Ce ratio. Within the first 32 min, manganese is predominantly dissolved, then (32–38 min) mostly the cerium dissolution is observed (Figure 5a), and from 43 to 46 min, the manganese is dissolved again. In the middle region (32–38 min), changes in the concentration of manganese and cerium during dissolution occur simultaneously. To illustrate this phenomenon, Appendix A display the dissolution curves of Mn and Ce normalized per the maximum dissolution rate. In the diapasons of 0–32 and 38–46 min, the Mn:Ce ratio varies from 0 to 8; especially large fluctuations are observed at the beginning of dissolution due to a very low concentration of Ce. From 32 to 38 min of dissolution, the temporal profile of the stoichiogram contains a close to linear segment; the Mn:Ce molar ratios change in the range of 0.24–0.5 (Figure 5a). The fragment of the stoichiogram can indicate the formation of concentration-inhomogeneous solid solutions or simultaneous dissolution of mixed oxide and another phase. From the stoichiogram, one can assume the phase composition with the minimum manganese content of Mn_0.24_Ce_1_ (without oxygen, because oxygen is not detected by DD). The first phase is dissolved within 5–40 min, its content being 77.9 wt.% (Figure 5b). After subtraction of the Mn_0.24_Ce_1_ phases from kinetic curves of Ce and Mn dissolution, the second and third phases are distinguished. The second phase contains only Mn in the amount of 21.9 wt.%, while the third phase contains Ce (0.2 wt.%) (Figure 5b). The dissolution of the Mn phase is observed in three regions: 0–32, 32–35 and 42–46 min (Figure 5b), which indicate the formation of different Mn oxides; this is in good agreement with XRD data (Table 1).

For the *Mn0.1Ce0.9* and *Mn0.15Ce0.85* catalysts, Appendix A display the dissolution curves of manganese and cerium as well as the stoichiograms of Mn:Ce versus time. The differential dissolution data also show the presence of a mixed Mn-Ce compound with the ratio Mn:Ce = 0.03:1 ÷ 0.06:1, which is much lower than that for *Mn0.45Ce0.55* (Mn:Ce = 0.24:1). The DD results show that *Mn0.1Ce0.9* contains 92.9 wt.% Mn_0.03_Ce_1_, 6.9 wt.% Mn_1_, and 0.2 wt%Ce_1_. For *Mn0.15Ce0.85,* 91.7 wt.% Mn_0.06_Ce_1_, 8.1 wt.% Mn_1_, and 0.2 wt.%Ce_1_ are observed. The DD results show that an increase in the manganese content in the catalyst leads to a growth in the content of free manganese (Mn_1_), which is associated with simple manganese oxides, and to an increase in the amount of manganese incorporated into the solid solution. From x = 0.1 to x = 0.45, the content of Mn oxides increases from 6.9 to 21.9 wt.%, while the substitution of Ce by Mn ions in the fluorite structure increases from Mn_0.03_Ce_0.97_O_2+δ_ to Mn_0.2_Ce_0.8_O_2+δ_.

For the *Mn0.45Ce0.55* catalyst, the DD, TEM and XRD data indicate that the thermal decomposition of oxalate precursors leads to the formation of a Mn-Ce solid solution. According to XRD, the lattice parameter and crystallite size differ from pure ceria. TEM shows agglomerates with the 2–4 nm nanoparticles containing Mn and Ce elements. EDS mapping indicates a uniform distribution of Mn and Ce cations over the particles. The DD data illustrate the joint dissolution of Mn and Ce atoms. 

Thus, the introduction of manganese into cerium oxide results in (i) a decrease in the crystalline size of ceria, (ii) a decrease in the surface area of the catalyst for x > 0.45; (iii) an increase in the content of simple oxides; and (iv) starting from x = 0.1–0.15, a solid solution Mn_y_Ce_1−y_O_2_ begins to form, which can be inhomogeneous. In the case of oxalate route, the formation of a solid solution may be determined by kinetic factors. At low x, there is a small number of contacts between oxalate precursors. Therefore, with a low content of manganese, a solid solution is either absent or formed in an insignificant amount, while at a higher concentration of Mn, a solid solution is formed.

### 3.2. Redox Properties

To understand the reducibility of the *MnxCe1−x* catalysts, TPR-H_2_ experiments were performed; all normalized results are shown in Figure 6. For the *Mn1Ce0* oxide, the TPR profile exhibits three reduction peaks with the maxima at 230, 320, and 430 °C. According to XRD, *Mn1Ce0* is a mixture of simple oxides such as Mn_5_O_8_, Mn_2_O_3_, and Mn_3_O_4._ The reduction of manganese oxides can be described by the successive processes Mn_5_O_8_ → Mn_2_O_3_ → Mn_3_O_4_ → MnO [44,45,46,47]. The pure CeO_2_ shows a single broad peak in a range of 300–560 °C, which can be attributed to the reduction of surface oxygen of ceria [48,49]. Modification with manganese cations influences the reducibility of ceria. The introduction of Mn into ceria results in shifting the TPR peak to low temperatures. As the manganese content increases to x = 0.25, the high-temperature peak shifts to T_max_ = 460 °C. For x > 0.25, apparently, there is an overlap of the reduction peaks of ceria and manganese oxides. 

For x = 0.1–0.15, intensive peaks with the maxima at ca. 410 °C and some hydrogen consumption at 270–320 °C appear, which are attributed to the reduction of manganese oxide. According to XRD data for the *MnxCe1−x* catalysts with x = 0.1–0.15, crystalline manganese oxides were not observed. However, DD shows a separate Mn-containing phase, which is in good agreement with the TPR data indicating hydrogen consumption due to the presence of highly dispersed manganese oxides. For x = 0.25, the TPR profile contains peaks with the maxima at 230, 270, 310, 410, and 460 °C. With increasing manganese content, the intensity of the peaks with the maxima at ca. 310–320 and 410–430 °C increases, which is associated with an increase in the content of simple manganese oxides. In addition, for x = 0.25–0.7, the reduction peak at 270 °C occurs. Zhang et al. [50] reported that hydrogen consumption in this area is attributed to the reduction of the Mn-Ce solid solution. However, according to other reports [26,38], two TPR peaks during the reduction of the Mn-Ce solid solution are also observed due to several reduction steps. In addition, the presence of amorphous manganese oxides makes it difficult to interpret the TPR data. For x = 0.7–0.8, an additional high-temperature peak with a maximum at 470 °C appears. It can be assumed that two high-temperature peaks (at 410 and 470 °C) correspond to the reduction of Mn_3_O_4_ to MnO, but from different precursors. This hypothesis is supported by the change in the intensity of these peaks with an increase in the manganese concentration. (For x = 0.6, hydrogen consumption during the peak at 410 °C is 1.65 mmol H_2_/g; for x = 0.7, the sum of intensities for peaks at 410 and 470 °C is 1.77 mmol H_2_/g; for x = 0.8, 2.00 mmol H_2_/g; and for x = 0.9, the intensity of peak at 415 °C is 2.66 mmol H_2_/g.) The separation of the reduction steps can be associated with the kinetics of the process and the influence of cerium on the reduction of manganese oxides. For x > 0.8, these two peaks overlap, and a broad peak located in the temperature range of 350–500 °C appears. With the addition of Mn from x = 0 to x = 1, a gradual increase in the amount of absorbed hydrogen from 0.44 to 5.7 mmol H_2_/g is observed. 

However, from TPR it is difficult to unambiguously ascribe the peaks of hydrogen consumption to the reduction steps of manganese and cerium cations, since the position of the peaks depends on many parameters such as dispersity, defectiveness, and presence of the second oxide. In the case of a multiphase system, the peaks often overlap. To study the reducibility of the *Mn0.45Ce0.55* catalyst, we have performed in situ XRD experiments.

### 3.3. In Situ XRD

The evolution of the phase composition of *Mn0.45Ce0.55* catalyst during heating under hydrogen flow is shown in Figure 7a (the corresponding XRD patterns are shown in Appendix A). According to in situ XRD data, the as-prepared catalyst contains not only ceria but also both the Mn_2_O_3_ and Mn_3_O_4_ phases. In the case of the in situ XRD experiment, the most intense diffraction lines of Mn_5_O_8_ were not observed due to the limited detector area for the in situ experiment and the phase was excluded from the calculation. All the phase transitions for manganese phases occur in the temperature range from 250 to 400 °C. At first, Mn_2_O_3_ is reduced to Mn_3_O_4_ starting from 260 °C and then Mn_3_O_4_ is further reduced to MnO starting from 320 °C. According to XRD data, the ceria content decreases from 86 to 62 wt.% in the same temperature region where the transformation of Mn oxides occurs. The observed behavior of the ceria content curve gives us a hint that manganese cations leave the fluorite structure with the formation of crystalline phases of manganese oxides at 250–400 °C. 

XRD data (Appendix A) show that a progressive decrease in the peak width of ceria due to thermal annealing is observed. In addition, no drastic crystallographic changes in ceria are observed, indicating that the rhombohedral Ce_7_O_12_, trigonal Ce_11_O_20_ or bixbyite phases reported by Bekheet [51] are not formed after treatment at 600 °C. Changes in the fluorite lattice upon treatment under H_2_ up to 600 °C are shown in Figure 7b. There are three regions of changes in the ceria lattice parameter. Up to 250 °C, the change in the lattice parameter is negligible; further heating leads to a rapid increase in the ceria lattice parameter in the temperature range of 250–400 °C. From 400 to 600 °C, the ceria lattice parameter rises almost linearly. The second region is noteworthy (from 250 to 400 °C), where the major changes in the lattice parameter (Figure 7b) and a decrease in the ceria content (Figure 7a) are observed. Chemical lattice expansion of ceria is mostly induced by the change in oxygen vacancy concentrations and the concomitant change of Ce^3+^ concentrations (for charge compensation) [51,52,53] or the reduction of Mn cations, although the segregation of the *MnCe* solid solution [53] and hydrogen incorporation into the lattice in the hydroxyl form [54,55] cannot be completely ruled out. In the case of the *Mn0.45Ce0.55* catalyst, the lattice parameter of a fluorite phase during the reduction increases from 5.402 Å at RT to 5.441 Å at 600 °C. During further cooling, a close-to-linear dependence was observed for the fluorite lattice parameter (Figure 7b). After the reduction, the lattice parameter does not return to the initial value and becomes 5.411 Å, which is close to that for pure ceria. Such a change in the lattice parameter simultaneously with a decrease in the ceria content indicates that during the reduction at 250–400 °C, the Mn cations leave the structure of the Mn_y_Ce_1−y_O_2−δ_ solid solution.

Interestingly, in situ XRD data shows that the reduction of Mn in the composition of the solid solution occurs within the same temperature ranges as the reduction of crystalline manganese oxides. Unfortunately, it was not possible to unambiguously attribute specific TPR peaks to the phase transformation observed by in situ XRD. According to in situ XRD, at 250–400 °C all Mn states are reduced to MnO. Whereas, according to TPR, the process is more extended in time and the reduction occurs up to 450 °C (Figure 6). Probably, differences in the temperature intervals of reduction are connected with different experimental conditions of in situ XRD and TPR method. For *Mn0.45Ce0.55,* in the temperature interval of 250–450 °C, there are only three TPR peaks with the maxima at 270, 320, and 410 °C. An increase in the Mn content from x = 0.45 to x = 0.7 leads to an increase in the content of crystalline manganese oxides and a decrease in the concentration of the solid solution (Table 1, Figure 6). At the same time, according to TPR, we observe a decrease in the peak at 270 °C and an increase in the intensity of the TPR peak at 320 °C (Figure 6). From this correlation, we can to some extent suppose that the TPR peak with a maximum at 270 °C corresponds to the reduction of solid solution, whereas the peak at 320 °C is attributed to the reduction of crystalline Mn_2_O_3_.

### 3.4. Surface Properties

To study the chemical states and relative concentrations of elements in the (sub)surface layers of catalysts, X-ray photoelectron spectroscopy (XPS) was used. Figure 8a shows the Ce3d core-level spectra of the catalysts. The shape of Ce*3d* spectra and the binding energy indicate that ceria in the catalysts exists in the Ce^3+^ and Ce^4+^ states. It is well known that the Ce*3d* core-level spectra of cerium oxide demonstrate the complex peak structure due to the presence of Ce^3+^ and Ce^4+^ states. Indeed, as a result of the spin-orbit interaction, the Ce*3d* core-level spectrum splits into two sublevels Ce*3d_5/2_* and Ce*3d_3/2_*, which leads to the appearance of a doublet in the XPS spectrum. Moreover, each component of the doublet, in turn, splits into three peaks in the case of CeO_2_ (v/u, v″/u″, v″′/u″′) or into two peaks in the case of Ce_2_O_3_ (v′/u′, v_0_/u_0_) [56,57]. According to the analysis of peak areas, the concentrations of Ce^3+^ cations for all the catalysts were calculated and the relevant data are presented in Table 2. 

Figure 8b shows the Mn2p core-level spectra of the catalysts. The Mn*2p* spectra are described by three Mn*2p_3/2_*–Mn*2p_1/2_* doublets assigned to manganese in the Mn^2+^, Mn^3+^ and Mn^4+^ states and by the corresponding shake-up satellites located at a distance of 7.8, 10.2 and 11.7 eV from the main peaks. In other words, manganese in the systems under consideration is in three states, Mn^2+^, Mn^3+^ and Mn^4+^. The Mn*2p_3/2_* peak assigned to manganese in the Mn^2+^ state lies at 640.0–640.5 eV; in the Mn^3+^ state, at 640.6–641.2 eV, and that of manganese in the Mn^4+^ state, at 641.2–642.2 eV. In the literature, for manganese in MnO, Mn_2_O_3_ and MnO_2_ oxides, the binding energies Mn*2p_3/2_* are given in the ranges of 640.4–641.7, 641.5–641.9, and 642.2–642.6 eV, respectively [58,59,60,61,62,63,64,65,66,67,68,69,70,71]. With an increase in the manganese content in the sample, the [Mn]/[Mn + Ce] atomic ratio and the amount of Mn in a higher oxidation state (Mn^2+^ and Mn^4+^) grow (Table 2). 

### 3.5. Catalytic Activity in CO Oxidation

Figure 9a shows the evolution of CO conversion as a function of temperature for *MnxCe1−x* catalysts. Pure CeO_2_ shows the lowest activity; the temperature of 50% CO conversion is 379 °C. The introduction of Mn results in a shift of the light-off curves toward low temperatures. An increase in the Mn content to x = 0.45 leads to a decline in the temperature of 50% CO conversion from 379 to 168 °C. A further increase in the manganese content leads to the opposite effect; the temperature of 50% conversion gradually grows up to 201 °C for *Mn0.9Ce0.1*. The rates of CO oxidation normalized per mass (R_1_) at 150 °C are presented in Figure 9b. One can see that the dependence of R_1_ versus x has a volcano shape and the *Mn0.45Ce0.55* catalyst exhibits the best activity. To elucidate the role of the amount of Mn ions, the rate of CO oxidation was normalized per the Mn content (R_2_ = R_1_/m(Mn)). From x = 0.1 to x = 0.25, the activity per the Mn content varies between 0.10 and 0.15 cm^3^/g(Mn)·s. For x = 0.45, R_2_ drastically rises up to 0.25 cm^3^/g(Mn)·s, and from x = 0.45 to x = 1 R_2_ decreases. Such a performance indicates that there is no linear dependence between activity and Mn content and that the origin of active sites changes depending on the composition. Similar dependencies for the catalysts were also found by other synthesis methods [50,72]. Appendix A shows the evolution of CO conversion with time over the *Mn0.45Ce0.55* catalyst.

## 4. Discussion

Next, we will discuss the results of catalyst characterization and catalytic tests with the purpose of identifying factors affecting the activity in CO oxidation reaction. When considering the surface area that determines active sites, one would expect that the activity will be governed by the surface area of the catalysts. Therefore, samples with the highest surface area should be the most active ones. However, although the *MnxCe1−x* catalyst with x = 0.1–0.45 possesses the highest surface area of ca. 100 m^2^/g (Table 1), its activity varies from 0.003 to 0.044 cm^3^/gs for R_1_ and from 0.10 to 0.25 cm^3^/g(Mn)s for R_2_. In the case of *MnxCe1−x* catalysts with x = 0.1–0.15, only the reflections of fluorite appear on the XRD patterns. TEM, DD, and TPR methods do not indicate a strong interaction between MnO_x_ and CeO_2_ with the formation of a solid solution. The addition of Mn ions (x = 0.1–0.15) does not lead to a drop in the surface area despite a decrease in the ceria amount. This indicates that MnO_x_ is in a highly dispersed amorphous state, but its activity is not high. With continuously increasing the Mn content, crystalline manganese oxides such as Mn_2_O_3_, Mn_3_O_4_, and Mn_5_O_8_ are formed and Mn ions incorporate into the structure of ceria. For x > 0.45, a decrease in the surface area is observed due to a growth in the content of crystalline Mn oxides. The R_1_ activity normalized per unit surface area increases from 4.4 10^−4^ to 5.7 10^−4^ cm^3^/m^2^s when x increases from 0.45 to 0.7, and then a decrease to 1.7 10^−4^ cm^3^/m^2^s is observed for x = 1. 

As our *MnxCe1−x* samples are composed of different phases, we also tried to find a correlation between a certain phase and catalytic activity. According to XRD, Mn-Ce catalysts contain crystalline oxides such as Mn_2_O_3_, Mn_5_O_8_, and Mn_3_O_4_ and ceria (the Mn-Ce oxide solid solution). An increase in the content of Mn oxides results in a decrease in the catalytic activity (Figure 9, Table 1). Appendix A shows the dependence of the ceria content estimated from XRD data on the manganese content (x). Up to x = 0.25, only ceria is observed; with an increase in the manganese content, its proportion gradually decreases. Simultaneously, the intensity of the surface content of Ce^3+^ decreases from 27% for x = 0–0.25 to 12–22% for x = 0.45–0.9. The generation of Ce^3+^ implies the creation of oxygen vacancy, based on the electroneutrality principle, which is thought to enhance the adsorption capacity of reactive oxygen species on the catalyst surface [12]. On the other hand, the incorporation of Mn^2+^ and Mn^3+^ also generates oxygen vacancies and could decrease the amount of Ce^3+^. The amount of oxygen vacancies depends on the degree of interaction between Mn and Ce oxides. Most authors [13,14,15,16,17,19,22,50,72] suppose that the formation of a solid solution is the key factor determining the activity of Mn-Ce catalysts. Our previous study [39] is in line with this; however, it was shown that, along with the formation of a solid solution, the transformations in manganese oxide nanoparticles could significantly improve the catalytic behavior. Herein, we did not observe such a tendency. According to the results of XPS, TEM, DD and the change in the structural characteristics of ceria, we speculate that a Ce_1−y_Mn_y_O_2_ solid solution begins to form starting from x = 0.25. Thus, the most active catalyst from the series of catalysts with varying manganese content is characterized by a balance of the following characteristics: the high specific surface area, the formation of a solid solution, and the maximum content of the solid solution.

## 5. Conclusions

A series of the *MnxCe1−x* oxide catalysts with a varying x from 0 to 1 has been prepared by an oxalate route with further calcination at 400 °C in air. Their catalytic performance was tested in CO oxidation. The structure, microstructure, redox and surface properties were studied by X-ray diffraction, high-resolution transmission electron microscopy with energy dispersive X-ray analysis, temperature-programmed reduction, X-ray photoelectron spectroscopy and the differential dissolution technique. The nanocrystalline oxides with a fluorite structure and particle sizes of 4–6 nm were formed. The introduction of manganese led to a reduction of the particle size of ceria and the formation of a Mn-Ce oxide solid solution based on the fluorite structure. An increase in the manganese content resulted in the formation of crystalline manganese oxides such as Mn_2_O_3_, Mn_3_O_4_, and Mn_5_O_8_. The catalytic activity of the Mn-Ce oxides in CO oxidation as a function of manganese content has a volcano-like shape, with the origin of active sites depending on the composition. At a low manganese content, MnO_x_ was in a highly dispersed amorphous state and provided the catalytic performance. The most active catalyst from the series of catalysts with varying manganese content is characterized by a balance of the following characteristics: the high specific surface area, the formation of a solid solution, and the maximum content of the solid solution. During a further increase in the manganese content, crystalline manganese oxides start to prevail, which leads to a drop in catalytic activity. 

## Figures and Tables

**Figure 1 nanomaterials-11-00988-f001:**
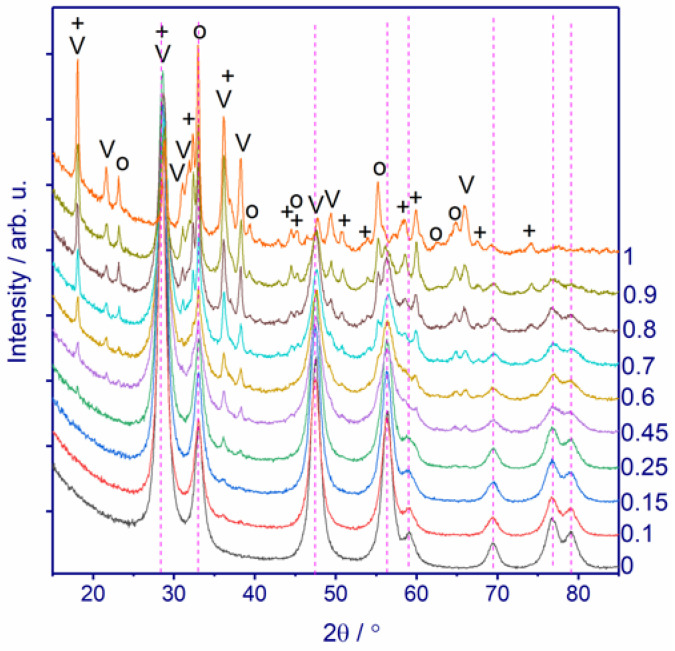
XRD patterns of *MnxCe1−x* (x = 0–1). The bar chart indicates the positions of CeO_2_ reflections, V–Mn_5_O_8_, o–Mn_2_O_3_, + –Mn_3_O_4_.

**Figure 2 nanomaterials-11-00988-f002:**
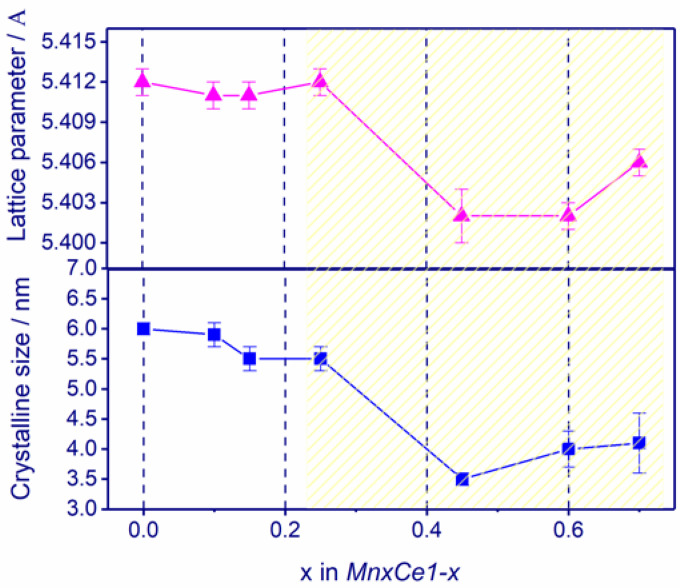
The lattice parameter and crystallite size and its dependence on the Mn content (*x*) in *MnxCe1−x.*

**Figure 3 nanomaterials-11-00988-f003:**
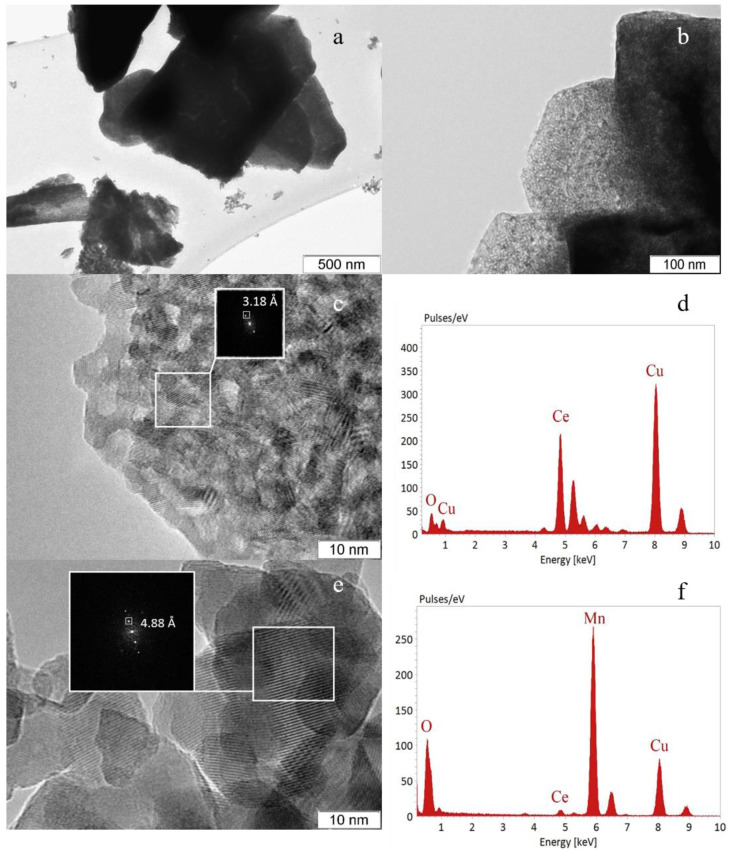
TEM images of *Mn0.15Ce0.85* (**a**–**c**,**e**) and EDX spectra of Ce-rich (**d**) and Mn-rich (**f**) regions.

**Figure 4 nanomaterials-11-00988-f004:**
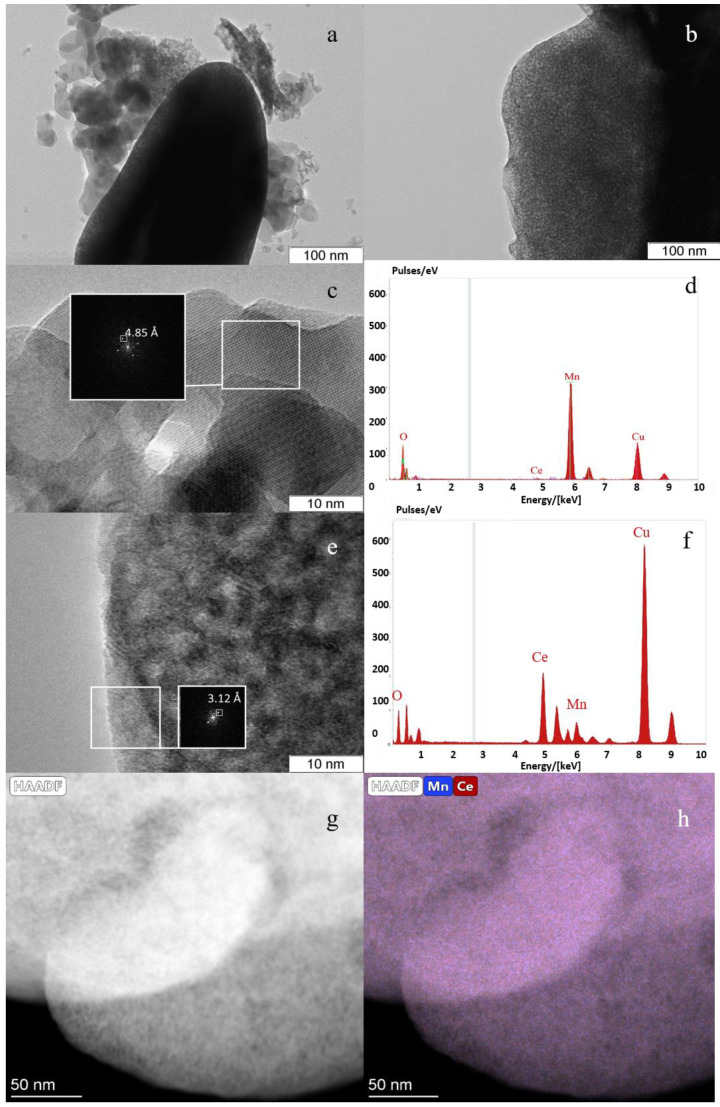
TEM images of *Mn0.45Ce0.55* (**a**–**c**,**e**), EDX spectra of Ce-rich (**f**) and Mn-rich (**e**) regions, EDS mapping (**g**,**h**).

**Figure 5 nanomaterials-11-00988-f005:**
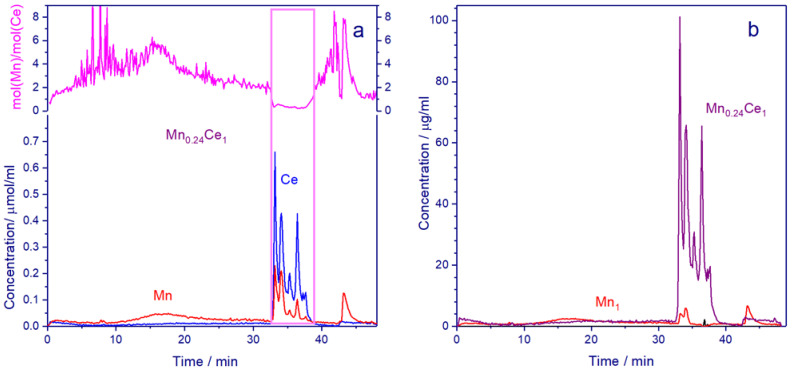
The dissolution curves for manganese and cerium, depending on time, the Mn/Ce ratio of the *Mn0.45Ce0.55* catalyst (**a**). The dissolution curves for Mn_1_ and Mn_0.24_Ce_1_ phases (**b**).

**Figure 6 nanomaterials-11-00988-f006:**
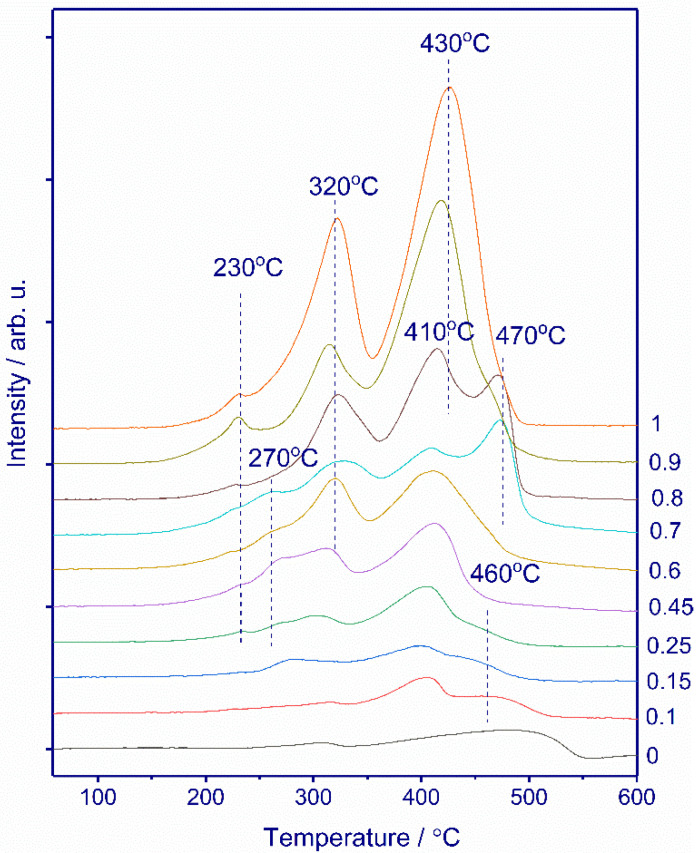
TPR-H_2_ profiles of *MnxCe1−x* catalysts.

**Figure 7 nanomaterials-11-00988-f007:**
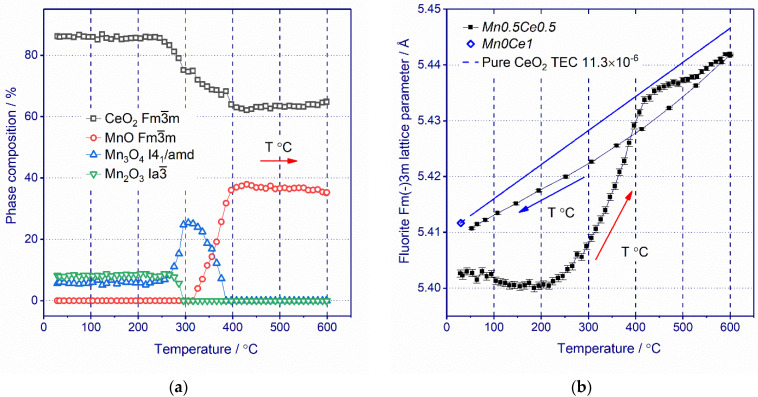
Changes in the fluorite cell parameter (**a**) and phase composition (**b**) during in situ reduction treatment under hydrogen for *Mn0.45Ce0.55* catalyst. The cell parameter *a* (blue diamond) for the as-prepared *Mn0Ce1* catalyst and its changes with temperature (blue dash line) considering TEC (α = 11.3 10^−6^ K^−1^ [41] is shown for comparison purposes).

**Figure 8 nanomaterials-11-00988-f008:**
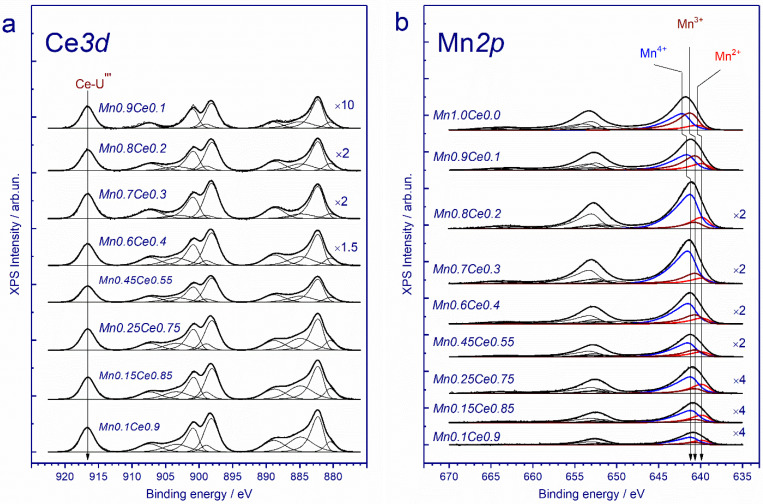
Ce*3d* (**a**) and Mn*2p* (**b**) core-level spectra of the catalysts under study. The spectra are normalized to the total intensity of the corresponding Ce*3d* and Mn*2p* core-level spectra.

**Figure 9 nanomaterials-11-00988-f009:**
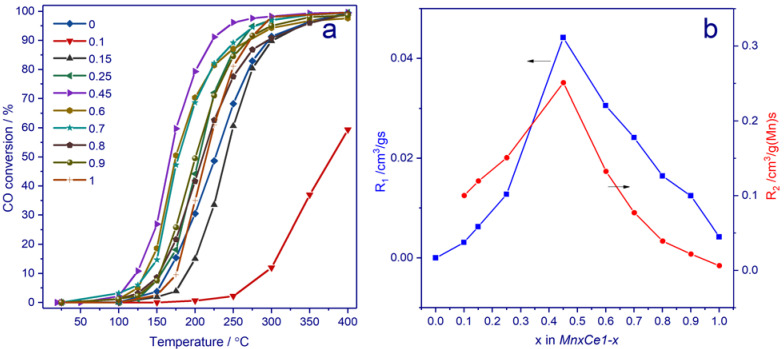
CO oxidation conversion (**a**) and CO oxidation rate at 150 °C normalized per gram (R1, blue dots) and Mn content (R2, red dots) (**b**) for *MnxCe1−x* catalysts.

**Table 1 nanomaterials-11-00988-t001:** Structural and microstructural characteristics of the *MnxCe1−x* catalysts.

Catalyst	Phase Composition, wt.%	Crystallite Size, nm	Lattice Parameter of Ceria, Å	BET Area, m^2^/g
*Mn0Ce1*	100% CeO_2_	6.0(1)	5.412(1)	110
*Mn0.1Ce0.9*	100% CeO_2_	5.9(2)	5.411(1)	99
*Mn0.15Ce0.85*	100% CeO_2_	5.5(2)	5.411(1)	99
*Mn0.25Ce0.75*	96% CeO_2_	5.5(2)	5.412(1)	100
	4% Mn_3_O_4_	25(3)		
*Mn0.45Ce0.55*	83% CeO_2_	3.5(1)	5.402(2)	102
	2% Mn_2_O_3_	-		
	7% Mn_3_O_4_	25(4)		
	8% Mn_5_O_8_	10(1)		
*Mn0.6Ce0.4*	72% CeO_2_	4.0(3)	5.402(1)	60
	4% Mn_2_O_3_	22(2)		
	17% Mn_3_O_4_	25(2)		
	7% Mn_5_O_8_	13(1)		
*Mn0.7Ce0.8*	62% CeO_2_	4.1(5)	5.406(1)	42
	8% Mn_2_O_3_	20(1)		
	12% Mn_3_O_4_	25(1)		
	18% Mn_5_O_8_	14(1)		
*Mn0.8Ce0.9*	45% CeO_2_	5.2(2)	-	31
	14% Mn_2_O_3_	20(1)		
	16% Mn_3_O_4_	25(1)		
	25% Mn_5_O_8_	13(1)		
*Mn0.9Ce0.1*	45% CeO_2_	5.2(2)	-	45
	14% Mn_2_O_3_	20(1)		
	16% Mn_3_O_4_	25(1)		
	25% Mn_5_O_8_	13(1)		
*Mn1Ce0*	28% Mn_2_O_3_	17(1)	-	32
	34% Mn_3_O_4_	25(1)		
	38% Mn_5_O_8_	13(1)		

**Table 2 nanomaterials-11-00988-t002:** Relative atomic concentrations of Mn and Ce cations in the near-surface layer of the studied catalysts.

Catalyst	Mn*2p_3/2_*			[Mn]/[Mn + Ce]	[Ce^3+^]/[Ce^3+^+Ce^4+^], %
	Mn^2+^ 639.9 eV	Mn^3+^ 640.7 eV	Mn^4+^ 641.4 eV		
*Mn0.1Ce0.9*	28	18	54	0.08	27
*Mn0.15Ce0.85*	28	15	57	0.14	28
*Mn0.25Ce0.75*	26	16	58	0.19	27
*Mn0.45Ce0.55*	18	27	55	0.41	21
*Mn0.6Ce0.4*	16	27	57	0.46	22
*Mn0.7Ce0.8*	11	22	67	0.59	12
*Mn0.8Ce0.9*	19	16	65	0.61	19
*Mn0.9Ce0.1*	18	39	43	0.94	17
*Mn1Ce0*	13	43	44	1.00	–

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
