# Peer review of "The Formation of Mn-Ce Oxide Catalysts for CO Oxidation by Oxalate Route: The Role of Manganese Content"

_nanomaterials, 2021, doi:10.3390/nano11040988_

Round 1

Reviewer 1 Report

Manuscript ID: nanomaterials-1164288

Authors : Olga A. Bulavchenko et al.

The submitted manuscript is similar to the article that authors published in Catalysis Letters 2021 (Reference [73] in this manuscript).

The authors declare at the end (on L 488 ): ”Our previous study [73] are in line with them, however it was shown that along with the  formation of a solid solution, the transformations in manganese oxide nanoparticles can significantly improve the catalytic behavior ….”.

In my opinion, with regard to the recurring facts from the publication [73], the article is very extensive and could be shortened.

In any case, I would expect reference [73] in the Introduction section

The goal as a reason for the writing this article should be given to L68.

The design of the Fig 5, Fig 8 and Fig 9 should be as the others (axis descriptions, font size).

Author Response

Response to Review

We are thankful to the refere for useful comments. We made revision of our manuscript according to the recommendations.

Referee 1

The submitted manuscript is similar to the article that authors published in Catalysis Letters 2021 (Reference [73] in this manuscript).

The authors declare at the end (on L 488 ): ”Our previous study [73] are in line with them, however it was shown that along with the  formation of a solid solution, the transformations in manganese oxide nanoparticles can significantly improve the catalytic behavior ….”.

In my opinion, with regard to the recurring facts from the publication [73], the article is very extensive and could be shortened.

In any case, I would expect reference [73] in the Introduction section

The goal as a reason for the writing this article should be given to L68.

Indeed, this article and publication in Catalysis Letters 2021 describe studies of Mn-Ce catalysts prepared by the oxalate method. Probably, articles look similar, since a similar set of methods are used (XRD, DD, XPS, TEM, TPR). However, current article discusses the role of the Mn / Ce ratio on the catalytic and physicochemical properties of the catalyst. In this case, we varied Mn content, and all catalysts were prepared under the same conditions and calcination was used at 400 ° C for 4 hours in air. In the article [Catalysis Letters 2021], on the contrary, we fixed the Mn: Ce ratio, and varied the annealing conditions (environment, heating rate, temperature, calcination time).

According to review’s recommendation, the reference [Catalysis Letters 2021] and goal of current study were added into Introduction section. «Previously, we studied the influence of the annealing conditions on the catalytic, structural, microstructural and redox properties of the Mn-Ce oxide catalysts obtained by thermal decomposition of oxalate precursors [39]. The goal of current study is to determine the effect of Mn content in Mn-Ce oxide catalysts on the catalyst properties.»

The design of the Fig 5, Fig 8 and Fig 9 should be as the others (axis descriptions)

According to review’s recommendation, we have changed the Figures 5,8,9.

Finally, we thank peer reviewer for attentive reading of the manuscript and for comments and suggestions, which helped us to improve the clarity of our manuscript.

Reviewer 2 Report

The paper presents a very interesting data about a mixed oxide catalyst for CO oxidation.

In my opinion the work is clear and well presented and a very wide characterization is also present.

I have just some minor corrections as follows:

  • The results section starts with this sentence:

This section may be divided by subheadings. It should provide a concise and precise 170 description of the experimental results, their interpretation, as well as the experimental 171 conclusions that can be drawn.

I think this is a part of the author instructions. Please remove it!

  • Captions of figure 3 and 4 should be rewritten indicating a description for each panel.

Thus, in my opinion, the work should be accepted after some minor revision.

Author Response

Response to Review

We are thankful to the referee for useful comments. We made revision of our manuscript according to the recommendations.

Referee 2

The paper presents a very interesting data about a mixed oxide catalyst for CO oxidation.

In my opinion the work is clear and well presented and a very wide characterization is also present.

I have just some minor corrections as follows:

The results section starts with this sentence:

This section may be divided by subheadings. It should provide a concise and precise 170 description of the experimental results, their interpretation, as well as the experimental 171 conclusions that can be drawn.

I think this is a part of the author instructions. Please remove it!

According to review’s recommendation, we have removed this part.

Captions of figure 3 and 4 should be rewritten indicating a description for each panel.

Captures of figure 3 and 4 were rewritten.

Thus, in my opinion, the work should be accepted after some minor revision.

We have checked the manuscript and corrected errors.

Finally, we thank peer reviewer for attentive reading of the manuscript and for comments and suggestions, which helped us to improve the clarity of our manuscript.

Round 2

Reviewer 1 Report

The manuscript has been modified and improved.

Therefore, I recommend it for acceptance in the present form. 

Reviewer 2 Report

The requested corrections have been done, so the manuscript can be accepted in the present form.